# HIF-1α Expression Increases Preoperative Concurrent Chemoradiotherapy Resistance in Hyperglycemic Rectal Cancer

**DOI:** 10.3390/cancers14164053

**Published:** 2022-08-22

**Authors:** Yi-Jung Huang, Yi-Ting Chen, Chun-Ming Huang, Shih-Hsun Kuo, Yan-You Liao, Wun-Ya Jhang, Shuo-Hung Wang, Chien-Chih Ke, Yu-Hsiang Huang, Chiu-Min Cheng, Ming-Yii Huang, Chih-Hung Chuang

**Affiliations:** 1Graduate Institute of Medicine, College of Medicine, Kaohsiung Medical University, Kaohsiung 80708, Taiwan; 2Drug Development and Value Creation Research Center, Kaohsiung Medical University, Kaohsiung 80708, Taiwan; 3Department of Pathology, Kaohsiung Medical University Hospital, Kaohsiung Medical University, Kaohsiung 80708, Taiwan; 4Department of Pathology, School of Medicine, College of Medicine, Kaohsiung Medical University, Kaohsiung 80708, Taiwan; 5Department of Radiation Oncology, Kaohsiung Medical University Hospital, Kaohsiung Medical University, Kaohsiung 80708, Taiwan; 6Department of Radiation Oncology, School of Medicine, College of Medicine, Kaohsiung Medical University, Kaohsiung 80708, Taiwan; 7Center for Cancer Research, Kaohsiung Medical University, Kaohsiung 80708, Taiwan; 8Department of Radiation Oncology, Kaohsiung Municipal Ta-Tung Hospital, Kaohsiung Medical University, Kaohsiung 80708, Taiwan; 9School of Medicine, College of Medicine, Kaohsiung Medical University, Kaohsiung 80708, Taiwan; 10Department of Medical Laboratory Science and Biotechnology, Kaohsiung Medical University, Kaohsiung 80708, Taiwan; 11Department of Medical Imaging and Radiological Sciences, Kaohsiung Medical University, Kaohsiung 80708, Taiwan; 12Department of Medical Research, Kaohsiung Medical University Hospital, Kaohsiung 80708, Taiwan; 13Post-graduate Year Training (PGY), Department of Clinical Education and Training, Kaohsiung Medical University Hospital, Kaohsiung 80708, Taiwan; 14Department and Graduate Institute of Aquaculture, National Kaohsiung University of Science and Techology, Kaohsiung 80708, Taiwan

**Keywords:** hyperglycemia, HIF-1α, CCRT, rectal cancer, HIF-1α inhibitors, concurrent chemoradiotherapy (CCRT), glycosylated hemoglobin, HbA1c, hypoxia-inducible factor-1 alpha (HIF-1α), glucose transport 1 (GLUT1), O-GlcNAc transferase (OGT)

## Abstract

**Simple Summary:**

Preoperative Concurrent Chemoradiotherapy (CCRT) is one of the standard treatments for patients with locally advanced rectal cancer. However, the efficacy of CCRT in hyperglycemic rectal cancer patients does not seem to be as expected. Therefore, we used clinical specimens, ectopic hyperglycemia animal models, and rectal cancer cells in a high-glucose environment to confirm how the high-glucose environment causes radiation resistance. The results confirmed that the high-glycemic environment could induce the overexpression of HIF-1α to cause CCRT tolerance in rectal cancer and suggest that combining HIF-1α inhibitors could reverse radioresistance in a high glucose environment. In future, HIF-1α inhibitors might be used as sensitizers to improve the efficacy of hyperglycemic rectal cancer patients receiving CCRT.

**Abstract:**

Purpose: Preoperative concurrent chemoradiotherapy (CCRT) is the standard treatment for locally advanced rectal cancer patients. However, the poor therapeutic efficacy of CCRT was found in rectal cancer patients with hyperglycemia. This study investigated how hyperglycemia affects radiochemotherapy resistance in rectal cancer. Methods and Materials: We analyzed the correlation between prognosis indexes with hypoxia-inducible factor-1 alpha (HIF-1α) in rectal cancer patients with preoperative CCRT. In vitro, we investigated the effect of different concentrated glucose of environments on the radiation tolerance of rectal cancers. Further, we analyzed the combined HIF-1α inhibitor with radiation therapy in hyperglycemic rectal cancers. Results: The prognosis indexes of euglycemic or hyperglycemic rectal cancer patients after receiving CCRT treatment were investigated. The hyperglycemic rectal cancer patients (*n* = 13, glycosylated hemoglobin, HbA1c > 6.5%) had poorer prognosis indexes. In addition, a positive correlation was observed between HIF-1α expression and HbA1c levels (*p* = 0.046). Therefore, it is very important to clarify the relationship between HIF-1α and poor response in patients with hyperglycemia receiving pre-operative CCRT. Under a high glucose environment, rectal cancer cells express higher levels of glucose transport 1 (GLUT1), O-GlcNAc transferase (OGT), and HIF-1α, suggesting that the high glucose environment might stimulate HIF-1α expression through the GLUT1-OGT-HIF-1α pathway promoting tolerance to Fluorouracil (5-FU) and radiation. In the hyperglycemic rectal cancer animal model, rectal cancer cells confirmed that radiation exposure reduces apoptosis by overexpressing HIF-1α. Combining HIF-1α inhibitors was able to reverse radioresistance in a high glucose environment. Lower HIF-1α levels increased DNA damage in tumors leading to apoptosis. Conclusions: The findings here show that hyperglycemia induces the expression of GLUT1, OGT, and HIF-1α to cause CCRT tolerance in rectal cancer and suggest that combining HIF-1α inhibitors could reverse radioresistance in a high glucose environment. HIF-1α inhibitors may be useful for development as CCRT sensitizers in patients with hyperglycemic rectal cancer.

## 1. Introduction

According to statistics from 2009 to 2013, about 16% of rectal cancer patients have hyperglycemia, and the number is continuing to rise every year [1,2]. It has been reported that hyperglycemia increased the risk of rectal cancer 1.3 times [3,4]. In addition, the mortality rate of rectal cancer patients with hyperglycemia is 17% higher than that of rectal cancer patients with euglycemia. In terms of the five-year survival rate, the overall survival rates of hyperglycemic colon cancer and rectal cancer were reduced by 18% and 19%, respectively, compared with euglycemia [5,6]. Therefore, more medical and care options for hyperglycemic rectal cancer patients are important [7,8,9,10]. A recent large-scale study of rectal cancer patients undergoing colectomy showed that the incidence of anastomotic leakage in hyperglycemic rectal cancer patients is four times that of euglycemic rectal cancer patients. Hyperglycemia also increases the risk of postoperative urinary tract infection leading to a prolonged hospital stay for postoperative patients and increases the risk of death in hospital [11,12,13,14]. Overall, hyperglycemic rectal cancer patients undergoing surgery have a higher risk of postoperative care and infection than euglycemic rectal cancer patients [15,16,17]. The use of chemotherapy or radiotherapy in hyperglycemic rectal cancer patients holds the possibility of having a better therapeutic effect than surgery.

In a study of hyperglycemic rectal cancer patients receiving chemotherapy, hyperglycemia significantly enhanced oxaliplatin chemotherapy resistance in patients receiving adjuvant infusional fluorouracil, leucovorin, and oxaliplatin (FOLFOX6) chemotherapy for stage III rectal cancer [18,19,20]. Hyperglycemia also affected the anti-tumor activity and survival rate of tumor-bearing or fibrosarcoma-bearing mice receiving oxaliplatin and 5-FU [21,22]. Therefore, a high glucose environment not only makes rectal cancer resistant to chemotherapy drugs but also aggravates the symptoms caused by hyperglycemia [23,24,25,26,27,28]. Using chemotherapy for hyperglycemic rectal cancer patients is a complex challenge. On the premise of reducing the risk of surgery and chemotherapy, preoperative concurrent chemoradiotherapy (CCRT) may be a better treatment option for patients with hyperglycemic rectal cancer. Preoperative CCRT is one of the standard treatments for patients with locally advanced rectal cancer. CCRT mainly uses radiotherapy combined with chemotherapy as a treatment method for prostate cancer and rectal cancer [29,30,31,32]. Its advantage is that it can reduce the staging of rectal cancer and preserve the anal sphincter.

Although CCRT has many advantages in the treatment of colorectal cancer patients, we observed that hyperglycemic rectal cancer patients undergoing CCRT seem to exhibit poor therapeutic efficacy. In this study, we wanted to identify the reasons for CCRT and radiotherapy resistance of rectal cancer in a high glucose environment. We compared the efficiency of CCRT in hyperglycemic rectal cancer and euglycemic colorectal cancer and analyzed the expression of HIF-1α in tumors to define the relationship between them. We detected the HIF-1α activation pathway of rectal cancer cells in different glucose concentrations and investigated whether the environment could affect the sensitivity of rectal cancer cells to 5-FU or radiation. Finally, we looked at whether inhibition of HIF-1α could reduce hyperglycemia and cause the radiation tolerance of rectal cancers. We further clarified the effects of the hyperglycemic environment and HIF-1α expression on CCRT resistance in rectal cancers.

## 2. Materials and Methods

### 2.1. Patients and Treatment Characteristics

The study included 60 rectal cancer patients receiving preoperative CCRT who had received curative resection at Kaohsiung Medical University Hospital (Kaohsiung, Taiwan) from January 2018 to December 2018. Patients with pre-operative distant metastasis, synchronous or metachronous malignancy or without complete medical records were excluded. Furthermore, 6 patients without available pre-CCRT specimens for analysis were excluded. The population was divided into two groups with high glycosylated hemoglobin (HbA1c > 6.5%) and low glycosylated hemoglobin (HbA1c ≤ 6.5%). The time interval between the end or CCRT and surgery was at least six weeks. The present study was approved by the ethical and research committee of Kaohsiung Medical University Hospital [approval no. KMUHIRB-E(I)-20210004]. All patients provided written informed consent for the addition of their sample to the collection, use for scientific research and added to a specimen bank.

The population with high glycosylated hemoglobin has taken antidiabetic drugs for at least three months and received CCRT for at least six weeks. At the beginning, patients underwent colonoscopy examination and an abdominal computed tomography study to identify the clinical stage according to the American Joint Committee on Cancer (AJCC) staging system. Patients at clinical T3-4 or N1-2 stage received preoperative CCRT (a combination of a long course of radiation therapy at doses of 45–50 Gy with 5-fluorouracil) and then the operative procedures were conducted according to previously published literature [33].

### 2.2. Pathologic Evaluation, Immunohistochemical Staining, and Tunel Assay

All samples were fixed in 4% formaldehyde for at least 6 h, embedded in paraffin, and then 4 μm-thick sections were cut. Hematoxylin and eosin slides of samples were reviewed to confirm the diagnosis and pathologic features, including tumor invasion status (lymphovascular or perineural), disease stage, and tumor regression score. Tumor regression score was used to evaluate the response of CCRT, according to the grading system of the College of American Pathologists. A four-grade scale is recommended and defined as: 0, no viable cancer cells (complete response); 1, single cells or rare small groups of cancer cells (near complete response); 2, residual cancer with evident tumor regression, but more than single cells or rare small groups of cancer cells (partial response); and 3, extensive residual cancer with no evident tumor regression (poor or no response). For immunohistochemical study slides, the sections were dried, deparaffinized, and then rehydrated. Antigen retrieval was performed using Heat-mediated Target Antigen Retrieval Buffer (pH 9.0; Dako, Glostrup, Denmark) for 8 min. The slides were washed with Tris buffer solution after using 3% hydrogen peroxide to block endogenous peroxidase activity for 5 min at room temperature. The sections were incubated with HIF-1α (1:500, Cat No. NB100-105SS, NOVUS) for 2 h at room temperature. Positive and negative controls were included for quality control. Low immunoreactivity of HIF-1α was defined as less than 50% positive cells with staining, and more than or equal to 50% positive cells was regarded as high HIF-1α expression. Tumor xenografts were surgically excised on day 30, placed in formalin, and embedded in paraffin. Tunel assay was conducted using In Situ Cell Death Detection Kit,POD (Lafayette, LA, USA) and quantified via image J (ImageJ bundled with 64-bit Java 1.8.0_172, Wayne Rasband, Bethesda, Rockville, MD, USA, https://imagej.net/imagej-wiki-static/Downloads (accessed on 28 June 2022)). All sections were scanned using the Olympus VS200 slide scanner (Olympus Soft Imaging Solutions Gmbh, Hamburg, Germany).

### 2.3. Cell Line and Culture 

HCT116 and SW480 colon cancer cell lines (Bioresource Collection and Research Center, Hsinchu 300193, Taiwan) were cultured in DMEM (Sigma-Aldrich, Darmstadt, Germany) supplemented with 10% bovine calf serum, 100 units/mL penicillin, and 100 μg/mL streptomycin at 37°C in an atmosphere of 5% CO_2_. High glucose medium was with 25 mM glucose and low glucose medium was with 5 mM glucose.

### 2.4. Western Blotting

Protein extracts, separated by SDS-PAGE and transferred onto NC membranes (Schleicher and Schuell, Einbeck, Germany), were probed with antibodies against GLUT1 (sc-377228, 1:1000, Santa Cruz, Heidelberg, Germany), OGA (JG40-05, 1:1000, Thermo Fisher, Branchburg, NJ, USA), OGT (D1D8Q, 1:1000, Cell Signaling Technology, Danvers, MA, USA), HIF-1α (GTX127309, 1:1000, GeneTex, Hsinchu, Taiwan), Hydroxy-HIF-1α (Pro564) (D43B5) Rabbit mAb (#3434, 1:1000, XP) or Anti-beta Actin antibody (mAbcam 8226, 1:1000, Abcam, Cambridge, UK) or proteins of interest were detected with peroxidase-conjugated AffiniPure Goat Anti-Mouse IgG, Fc Fragment Specific (115-035-008, 1:1000, Jackson ImmunoResearch, West Grove, PA, USA) or Rabbit Anti-Goat IgG Antibody, HRP conjugate, (abbbit IgG, AP106P, 1:1000, Sigma-Aldrich) and visualized with the Immobilon Western Chemiluminescent HRP Substrate (WBKLS0500, Merck, Rahway, NJ, USA), according to the protocol provided. The Western blotting was repeated three times and included in the statistics. Quantitation of bands was conducted using Image J.

### 2.5. Irradiation Exposure and Cell Viability Assay

Cells were plated in 96-well plates (2000/well) overnight irradiated at 2 Gy per session by a linear accelerator (Varian Clinac iX, Department of Radiation Oncology, Kaohsiung Medical University Hospital) a bolus (1.5 cm-thick) was placed on the top and bottom of cell culture plates. All cell viability was detected using the ATPlite kit (510-17281, PerkinElmer, Chennai, India) and the luminesce value was measured using a multimode plate reader (VICTORTM X2, PerkinElmer, Chennai, India).

### 2.6. Colony Formation Assay

Cells (10^3^ cells/well) were seeded in 6-well dishes in medium after treating cells with/without LW6 or ionizing radiation expose after culture for three weeks. The media was gently removed from each of the plates by aspiration, then washed with 5 mL 0.9% saline. The colonies were fixed with 5 mL 10% neutral buffered formalin solution for 15–30 min and stained with 5 mL 0.01% (*w*/*v*) crystal violet in dH_2_O for 30–60 min. Excess crystal violet was washed with dH_2_O and dishes were allowed to dry. Colonies containing more than 50 individual cells were counted using a stereomicroscope.

### 2.7. Establishment of Hyperglycemia Ectopic Rectal Cancer and Combined LW6 with Irradiation Treatment Schedule

Eight-week-old, male nude mice (BALB/cAnN.Cg-Foxn1nu/CrlNarl) from the National Laboratory Animal Center, Taiwan, were used. The average weight was 20–25 g, and the average blood glucose concentration nonfasting was 90–150 mg/dL. To establish an animal model of hyperglycemia ectopic rectal cancer, mice were fasted for four hours before injecting with Streptozotocin (50 mg/kg) for five days through intraperitoneal injection, after detecting blood glucose concentration to confirm the establishment of hyperglycemic mice (nonfasted blood glucose concentration was >200 mg/dL). The HCT116 and SW480 cell suspension (2 × 10^6^ in PBS) was subcutaneously inoculated into the right hind leg of euglycemic and hyperglycemic mice. When tumors reached a size of approximately 50 mm^3^, radiation was conducted with multiple doses of 2 Gy per fraction (cumulative radiation dose was 6 Gy) by the linear accelerator before the bolus was placed on the tumor top of mice. The mice were treated with or without LW6 (4 mg/kg) by intraperitoneal injection every three days; the body weight (g) and blood sugars (mg/dL) were measured every two days until the experiment endpoint (day 30). The mice were sacrificed and the tumors were collected to observe at day 30. After sacrificing the mice, the tumor mass was removed and analyzed for HIF-1α expression and Tunnel assay by IHC. The tumor size was measured using a digital caliper (VWR International, Radnor, PA, USA), and the tumor volume was determined using the formula: tumor volume [mm^3^] = (length [2]) × (width [2])^2^ × 0.52. The present study was approved by Kaohsiung Medical University Institutional Animal Care and Use Committee [approval no. 109224]. All experimental procedures complied with regulations.

### 2.8. Statistical Analysis 

The Chi-square test was used in Table 1 and Table 2 to determine HbA1c status and HIF-1α protein immunohistochemical expression in comparing pre-CCRT specimens correlated with clinical-pathological factors. IHC stain areas are quantified, and the Western blot method was analyzed using image J. *T*-test was used in ATPlite, Colony formation, and tumor volume analysis. The T-test was two-tailed. The data are expressed as mean ± standard deviation (SD). The statistical significance level was *p* < 0.05.

## 3. Results

### 3.1. Hyperglycemia Affects the Efficacy of CCRT in Patients with Rectal Cancer

In order to investigate the effect of hyperglycemia on rectal cancer patients receiving CCRT, 54 rectal cancer patients who had preoperative CCRT were collected and divided into two groups based on glycosylated hemoglobin (HbA1c) level. Hyperglycemia was identified as HbA1c greater than 6.5% (*n* = 13, HbA1c > 6.5%), and HbA1c less than or equal to 6.5% was regarded as euglycemia (*n* = 41, HbA1c ≤ 6.5%). Table 1 shows the HbA1c level in relation to patient characteristics. The results revealed that HbA1c differed significantly according to pathologic tumor disease stage (*p* = 0.012), tumor regression score (*p* = 0.017), and pathologic complete response (*p* = 0.047). Patients with hyperglycemic status had higher tumor regression score and less pathologic complete response. The above findings indicate that rectal cancer patients with hyperglycemia have a poorer response than the ones with euglycemia after receiving CCRT treatment.

### 3.2. HIF-1α Protein Positively Correlates with Tumor Regression Score in Rectal Cancer Patients Receiving CCRT Treatment

To explore the reasons for the poor response and inferior efficacy of pre-operative CCRT in rectal cancer patients with hyperglycemia, we performed HIF-1α immunohistochemical staining and scored it according to the degree of staining of patient’s tissue samples (Figure 1). The immunoreactivity of HIF-1α protein correlated with clinical-pathological characteristics was analyzed. The data showed high levels of HIF-1α expression in 17 (31.5%) and low levels in 37 (68.5%) of 54 pre-CCRT rectal cancer cases (Table 2). Immunoreactivity of HIF-1α significantly differed by lymphovascular invasion (*p* = 0.009), peri-neural invasion (*p* = 0.002), tumor regression score (*p* < 0.001), pathologic complete response (*p* = 0.01), and HbA1c level (*p* = 0.046). High HIF-1α was significantly associated with more lymphovascular and peri-neural invasion, higher tumor regression score, less complete response, and hyperglycemia. The above results indicated that a high glucose environment may affect the CCRT response associated with the HIF-1α signaling pathway in rectal cancer cells.

### 3.3. High-Glucose-Environment-Induced GLUT1-OGT-HIF-1α Signaling Pathways in Rectal Cancer Cell Lines

Ferrer et al. indicated that human breast cancers with high levels of HIF-1α contain elevated OGT, and lower OGA levels correlate independently with poor patient outcomes. This result suggests that OGA or OGT may be involved in high-sugar-environment regulation in rectal cancer. In order to confirm whether rectal cancer cells also activate this signaling pathway in a high glucose environment, human colorectal cancer cell lines HCT116 and SW480 were treated with different glucose concentrations of 5, 15, 25, and 35 mM, respectively, and cytoplasmic mass was collected after 48 h of reaction. The levels of GLUT1, OGA, OGT, HIF-1α, Hydroxy-HIF-1α (HIF-1α-OH), and β-actin were measured. The results for HCT116 or SW480 are shown in Figure 2, respectively. The GLUT1, OGT, and HIF-1α levels at glucose concentrations of 25 mM were higher than 5 mM in HCT116 and SW480. The OGA, HIF-1α-OH level at 5 mM glucose was lower than that at 25 mM in both HCT116 (Figure 2A,C) and SW480, (Figure 2B,D). These results suggested that the high-glucose environment promotes the OGT, HIF-1α, and GLUT1 signal pathways and reduces the level of OGA and HIF-1α-OH in HCT116 and SW480.

### 3.4. A High Glucose Environment Increases the 5-FU and Radiation Tolerance in Rectal Cancer Cell Lines

To confirm whether the high glucose environment affects the 5-FU and radiation tolerance of rectal cancer cells, HCT 116 or SW480 cells were cultured in 5, 15, or 25 mM glucose medium with 5-FU in serial dilution for 48 h, after analyzing cell viability. The results showed that the cell viability of 25 mM glucose medium (IC50 = 32.45 ± 6.77 μM) was higher than 15 mM glucose medium (IC 50 = 1.9 ± 0.516 μM) or 5 mM glucose medium (IC50 = 0.55 ± 0.03 μM) in HCT116 (Figure 3A). For SW480, the cell viability of 25 mM glucose medium (IC50 = 148.43 ± 12.33 μM) was higher than 15 mM glucose medium (IC50 = 28.06 ± 2.63 μM) or 5 mM glucose medium (IC50= 3.09 ± 0.33 μM) (Figure 3B). The above results indicate that a high glucose environment increases the 5-FU tolerance in rectal cancer cell lines. Furthermore, to clarify whether the high glucose environment affects the radiation resistance of rectal cancer, HCT 116 or SW480 were cultured in 5, 15, and 25 mM glucose medium, after exposure to 0, 2, or 6Gy for 48 h and then cell viability was analyzed. The results showed at 2Gy the cell viability means of 5, 15, or 25 mM glucose medium were 57.43 ± 7%, 78.25 ± 2.05%, or 86.53 ± 0.61%, respectively; at 6Gy radiation, the cell viability means of 5, 15, or 25 mM glucose medium were 21.63 ± 0.93%, 38.54 ± 0.22%, or 53.5 ± 2.18%, respectively, in HCT116 (Figure 3C). For SW480 at 2Gy, the cell viability means of 5, 15, or 25 mM glucose medium were 65.41 ± 4.56%, 88.75 ± 0.51%, and 98.08 ± 0.26%, respectively; at 6Gy radiation-exposure, the cell viability means of 5, 15, or 25 mM glucose medium were 20.39 ± 2.5%, 35.04 ± 3.63%, and 45.41 ± 2.05% (Figure 3D). These results suggest that both HCT116 and SW480 have radiation tolerance in a high sugar environment. In order to determine the cell radiation tolerance, we further analyzed cell proliferation through colony formation. HCT116 were cultured in 5, 15, or 25 mM glucose media and then exposed to 2Gy radiation after incubation for two weeks. The results showed the cell number of HCT116 means at 5, 15, or 25 mM glucose concentration were 3.33 ± 1.53, 56.67 ± 7.64, or 253.33 ± 7.64 (Figure 3E,G). In SW480, the cell number means at 5, 15, or 25 mM glucose concentrations were 13 ± 4.36, 139.67 ± 3.21, or 322.33 ± 52.39 (Figure 3F,H). The above results show that a high glucose environment causes radiation-resistance in HCT116 and SW480 cells.

### 3.5. Hyperglycemia Promotes the Radioresistance of Rectal Cancer Cells by Overexpression of HIF-1α

To confirm that hyperglycemia causes radiation resistance in rectal cancer, hyperglycemic ectopic rectal cancer animal models were established through subcutaneous injection of HCT116 or SW480 until the tumor volume reached 50 mm^3^. The tumor was treated with 6Gy in the first week and then the tumor volume was measured per three days. On the 30th day, the results indicated the tumor mean volume of HCT116-bearing hyperglycemic mice that received radiation exposure was 648.75 ± 25.94 mm^3^ and the tumor mean volume of HCT116-bearing euglycemic mice that received radiation was 359 ± 16.43 mm^3^ (Figure 4A, *p* = 0.004). In SW480-bearing mice, the tumor mean volume of hyperglycemic mice was 486 ± 72.54 mm^3^, and the tumor mean volume of hyperglycemic mice was 787.59 ± 74.59 mm^3^ on the 30th day (Figure 4B, *p* = 0.001). The above results suggest that hyperglycemia increased rectal cancer resistance to radiation. Further, to verify whether this phenomenon was caused by overexpression of HIF-1α, we analyzed the expression level of HIF-1α and the proportion of apoptosis by IHC stain and Tunnel assay in HCT116 tumors and SW480 tumors. The results indicated that hyperglycemic conditions resulted in higher levels of expression of HIF-1α than in euglycemic conditions in HCT116. The level of tumor cell apoptosis showed the opposite trend (Figure 4C,E). Furthermore, we analyzed the amount of expression of HIF-1α and the proportion of apoptosis in SW480. The results suggested that hyperglycemia resulted in a higher level of expression of HIF-1α than in euglycemic conditions. Tumor cell apoptosis level showed the opposite trend (Figure 4D,F). Based on the above results, we determined that hyperglycemia can induce radiation resistance via overexpression of HIF-1α in rectal cancer.

### 3.6. LW6 Reduces the Radiation Tolerance of Rectal Cancer via Inhibiting the Expression of HIF-1α in a High Glucose Environment

To confirm that the induction of HIF-1α in a hyperglycemic environment was the main cause of radiation resistance, we treated rectal cancer with LW6 (an HIF-1α inhibitor) and radiation in a high glucose environment. HCT116 and SW480 were treated with 10 μM LW6, radiation of 2 Gy, or a combination of 10 μM LW6 with radiation at 2 Gy for 48 h and cell proliferation was analyzed. The results showed colony formation (%) at 25 mM glucose concentration (90.41 ± 0.77%) was higher than at 5 mM (48.63 ± 3.86%) after radiation exposure (*p* = 0.004). However, addition of LW6 was able to reverse the radioresistance of HCT116 in a high glucose environment (*p* = 0.017), Colony formation of combination LW6 with radiation at 25 mM was 30.6 ± 3.1%, at 5 mM was 10.97 ± 4.56% (Figure 5A). For SW480, the colony formation at 25 mM glucose concentration (44.7 ± 0.87%) was higher than 5 mM (28.39 ± 0.91%) after radiation exposure (*p* = 0.003). However, addition of LW6 was able to reverse the radioresistance of SW480 in a high glucose environment (*p* = 0.003). Colony formation of combination LW6 with radiation at 25 mM was 1.41 ± 0.57%, at 5 mM was 14.47 ± 0.39% (Figure 5B). Moreover, the above results show that LW6 reduces the radiation tolerance of rectal cancer via inhibiting expression of HIF-1α in a high glucose environment. To further confirm the above phenomenon, whether HIF-1α is dominated by the amount of HIF-1α expression, we collected cell plates from HCT116 and SW480 treated without or with LW6 or radiation in 5 or 25 mM glucose medium and then detected the HIF-1α/β-actin ratio. The results show that the HIF-1α/β-actin ratio in 25 mM glucose medium, the combination LW6 with radiation is 0.2 ± 0.075, and the radiation alone is 0.62 ± 0.057 in HCT116 (Figure 5C,E). The HIF-1α/β-actin ratio is in 25 mM glucose medium, the combination of LW6 with radiation is 0.16 ± 0.068, and the radiation alone is 0.45 ± 0.096 in SW480 (Figure 5D,F).

### 3.7. LW6 Increases the Efficacy of Radiotherapy for Hyperglycemia Ectopic Colorectal Cancer by Inhibiting HIF-1α Expression

To confirm that LW6 can increase the effect of radiotherapy in hyperglycemia-colorectal cancer, we used the hyperglycemia ectopic colorectal cancer animal model treated with/without LW6, radiation, or combined LW6 inhibitor (4 mg/kg) with radiation, and measured the tumor size per three days until the 30th day. The mice were sacrificed and the tumors were collected for observation at day 30 (Appendix A).

The results showed that the volume of the combated radiation with LW6 (378 ± 43.52 mm^3^) was significantly smaller than the radiation alone (554.25 ± 38.91 mm^3^, *p* < 0.01), which means that LW6 can increase the therapeutic effect of radiation on HCT116 tumors (Figure 6A). The same result was also obtained in SW480 tumors (*p* < 0.01), the volume of the combated radiation with LW6 was 302.5 ± 55.72 mm^3^, the radiation alone was 712.75 ± 47.46 mm^3^ (Figure 6B). To understand the correlation therapeutic efficacy of radiation with HIF-1α level in rectal cancer, we analyzed the expressed HIF-1α level and apoptosis in HCT116 or SW480, respectively. The results suggested that the group treated with radiation with LW6 had a lower HIF-1α level than the group treated with radiation alone in HCT116. To analyze the apoptosis of the tumor, the results indicated combined radiation with LW6 showed more apoptosis than the group treated with radiation alone (Figure 6C,E). SW480 had the same results (Figure 6D,F). This means that LW6 can reduce radiation tolerance of rectal cancer in hyperglycemia by reducing HIF-1α level.

## 4. Discussion

In this study, we analyzed rectal cancer patients with euglycemia and hyperglycemia after radical CCRT treatment. We found that patients with hyperglycemia rectal cancer have a poor prognosis which positively correlates with HIF-1α expression. Although the findings confirm the association of hyperglycemia, HIF-1a, and CCRT efficacy in rectal cancer patients, the population of this study is too small and the number of cases received needs to increase to more accurately confirm the association between the factors. In addition, we confirmed that a high glucose environment could induce the GLUT1-OGT-HIF-1α signaling pathway to promote 5FU-resistance and radioresistance of rectal cancer cells by overexpressing HIF-1α. The in vivo results indicated that hyperglycemia could induce HIF-1α expression and cause rectal cancer radioresistance. Moreover, the HIF-1α inhibitor, LW6, could increase the efficacy of radiotherapy for hyperglycemia ectopic rectal cancer by inhibiting HIF-1α expression. In the future, it may be possible to use HIF-1α inhibitors to target CCRT in hyperglycemic rectal cancer patients.

Hyperglycemia promotes the progression and reduces therapeutic efficacy of tumors [34]. For example, Wenjie and colleagues pointed out that hyperglycemia can provide nutrition to malignant tumor cells to accelerate the rapid proliferation and progression of many types of tumor cells, including breast cancer, lung cancer, and rectal cancer [35]. Hyperglycemia has also been reported to possibly increase MMP expression to increase the invasive ability of colorectal cancer [36] and pancreatic cancer [37,38]. Ma and colleagues revealed that hyperglycemia weakened the anti-proliferative effect of 5-fluorouracil (5-FU) on colon cancer cells, and rectal cancer patients with hyperglycemia required higher doses of 5-FU and longer chemotherapy to adequately inhibit the growth of tumor cells [39,40]. This suggests that hyperglycemia also increases the chemo-resistance of cancer. In addition, clinical studies reported that patients of head and neck cancer with diabetes mellitus have more side effects when undergoing CCRT as compared with patients without diabetes mellitus, such as higher rates of infection, hematotoxicity, loss of body weight, and higher treatment-related mortality [41,42]. Here, our clinical research also suggests that rectal cancer patients with hyperglycemia that have a poor prognosis undergo CCRT as compared with euglycemia patients. Our in vitro and in vivo studies also confirmed that a high glucose environment could promote the 5FU-resistance and radioresistance of rectal cancer, demonstrating that hyperglycemia plays a very important role in the therapeutic efficacy of rectal cancer. According to the above findings, hyperglycemia stimulates tumor progression and reduces therapeutic efficacy. In this study, we only focused on the correlation of hyperglycemia with the prognosis of rectal cancer CCRT during the first three months of treatment, and hyperglycemia was also the main factor in cell and animal experiments. Therefore, we did not discuss the diabetic history of all the patients and anti-diabetes medication history. In the future, it may be an interesting topic to study the correlation of the prognosis of rectal cancer patients CCRT with/without receiving anti-diabetic drugs.

A high glucose environment increasing HIF-1α expression leads to radiotherapy resistance in cancer. In a related report, it was pointed out that in the hyperglycemic mouse model, the high glucose environment can increase the expression of HIF-1α through glucose-responsive carbohydrate response element binding protein (ChREBP), which leads to an increase in the expression of HIF-1α [43,44]. In addition, it was found in hyperglycemic patients that the hypoxia-inducible factor 1α–6-phosphofructo-2-kinase/fructose-2,6-bisphosphatase 3 (PFKFB3) pathway and activation of the PI3K/AKT pathway was induced to increase the expression of HIF-1α [45,46]. An animal model of pancreatic cancer with hyperglycemia showed that HIF-1α is abundantly expressed in cancer cells [47,48]. It has been suggested that overexpression of HIF-1α in many cancer cells results in resistance to chemotherapy, for example, Xia et al. suggested that HIF-1α inhibited the chemo-/radiotherapy-induced apoptosis of tumor cells in colon cancer through the promotion of antiapoptotic protein [49,50]. The above studies showed the correlation between HIF-1α expression and radiation resistance. In addition, a study of the correlation between carbohydrate metabolism and HIF-1α expression suggested that O-GlcNAcylation regulates glycolysis in cancer cells via HIF-1α [51,52]. Unfortunately, there is a lack of a hyperglycemic environment for radiation resistance. Here, we not only confirmed that patients with clinical hyperglycemia have a poor CCRT prognosis, but also confirmed that the expression level of HIF-1α in rectal cancer is higher than other euglycemic rectal cancer patients. We also confirmed that the high glucose environment activates the GLUT1-OGT-HIF-1α signal pathway causing 5-FU and radiation resistance, and hyperglycemia can induce HIF-1α expression by analyzing cancer specimens of mice with rectal cancer. Therefore, HIF-1α is a key CCRT resistance factor in hyperglycemic rectal cancer.

HIF-1α inhibition might be a potential solution to enhance the chemo-/radiation therapeutic efficacy for cancer. There are a number of HIF-1α inhibitors in development as cancer therapeutics, such as EZN-2968, Acriflavine, and LW6. A study by Enrica Borsi et al. indicated that EZN-2968 targets HIF-1α mRNA suppression to cause a metabolic shift that leads to increased production of ATP by oxidative phosphorylation and affected apoptosis to improve multiple myeloma treatment [53,54]. In addition, Mangraviti et al. also pointed out that Acriflavine as an HIF-1α inhibitor that can increase the therapeutic effect of radiation on brain cancer and prolong the survival rate of mice [55]. Zhang et al. stated that Metformin and LW6 impair pancreatic cancer cells via reducing nuclear localization of YAP1 [56]. The above studies pointed out that HIF-1α inhibitors have therapeutic effects in cancer. In this study, we confirmed that LW6 is an HIF-1α inhibitor for hyperglycemic rectal cancer. As shown in Figure 2, we found that the OH-HIF-1α decreases dependence on the increase in high sugar concentration, showing that a high sugar environment increases the stability of HIF-1α. LW6 potently inhibits HIF-1α accumulation by degrading HIF-1α without affecting the HIF-1α mRNA levels [57]. It was confirmed that inhibiting HIF-1α can increase the efficacy of radiation for hyperglycemic rectal cancer (Figure 5C,D). The mechanism may be that hyperglycemia impairs the function of HIF-1α inhibitors and weakens the resistance of HIF-1α inhibitors to tumor chemotherapy or radiotherapy. Overall, HIF-1α inhibitors were able to block the hyperglycemia-induced GLUT1-OGT-HIF-1α signaling pathway and reduce rectal cancer chemoradiation resistance. In conclusion, our study revealed a novel mechanism of hyperglycemia-dependent chemoresistance and provided a novel strategy for CCRT-resistant cancer therapy. In the future, perhaps HIF-1α inhibitors can be used as CCRT sensitizers in patients with hyperglycemic rectal cancer.

## Figures and Tables

**Figure 1 cancers-14-04053-f001:**
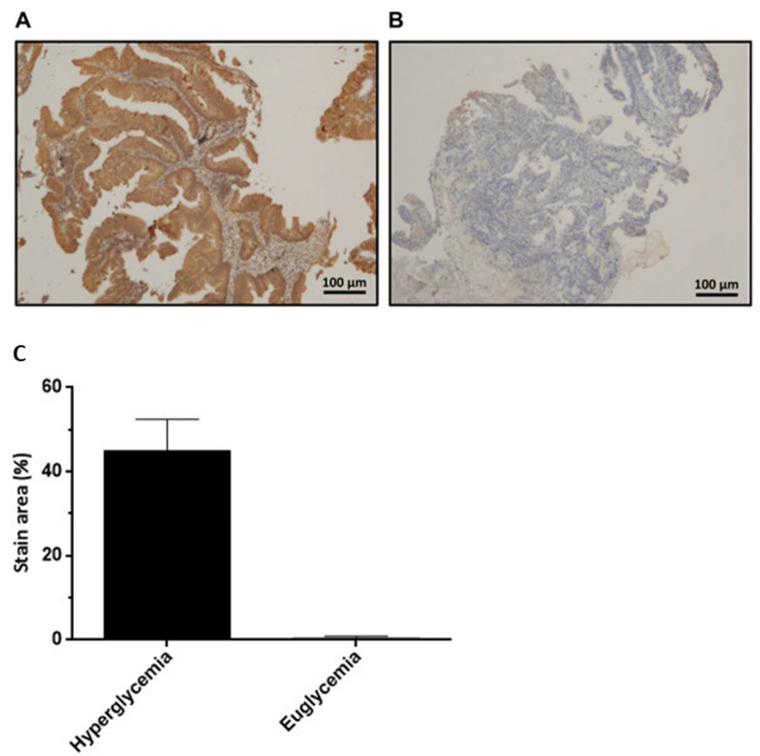
Representative immunohistochemical staining of HIF-1α in rectal cancer specimens. High (**A**) and low (**B**) levels of HIF-1α in rectal cancer specimens. Scale bar with microscope magnified 100 times. (**C**) The HIF-1α staining in rectal cancer specimens also quantitated by image J. soft.

**Figure 2 cancers-14-04053-f002:**
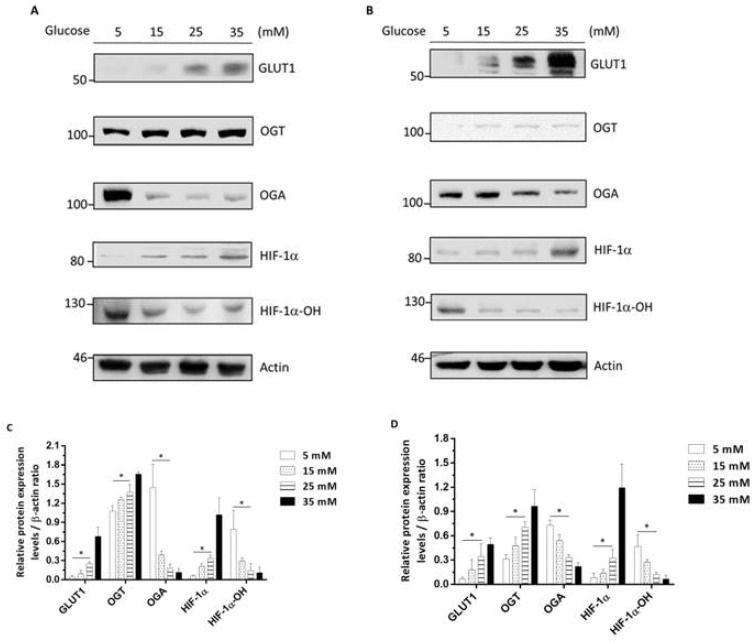
High glucose activates the GLUT1-OGT-HIF-1α signaling pathway leading to HIF-1α expression. HCT116 (**A**) and SW480 (**B**) cells were treated with 5, 15, 25, and 35 mM. After 48 h the cytoplasmic extrusion mass was collected and the endogenous protein levels of GLUT1, OGT, OGA, HIF-1α-OH, HIF-1α, and β-actin were evaluated using antibodies and Western blotting. Relative protein expressions in HCT116 (**C**) and SW480 (**D**) cells are shown. Data are presented as mean  ±  SD (*n*  =  3; * *p* < 0.05).

**Figure 3 cancers-14-04053-f003:**
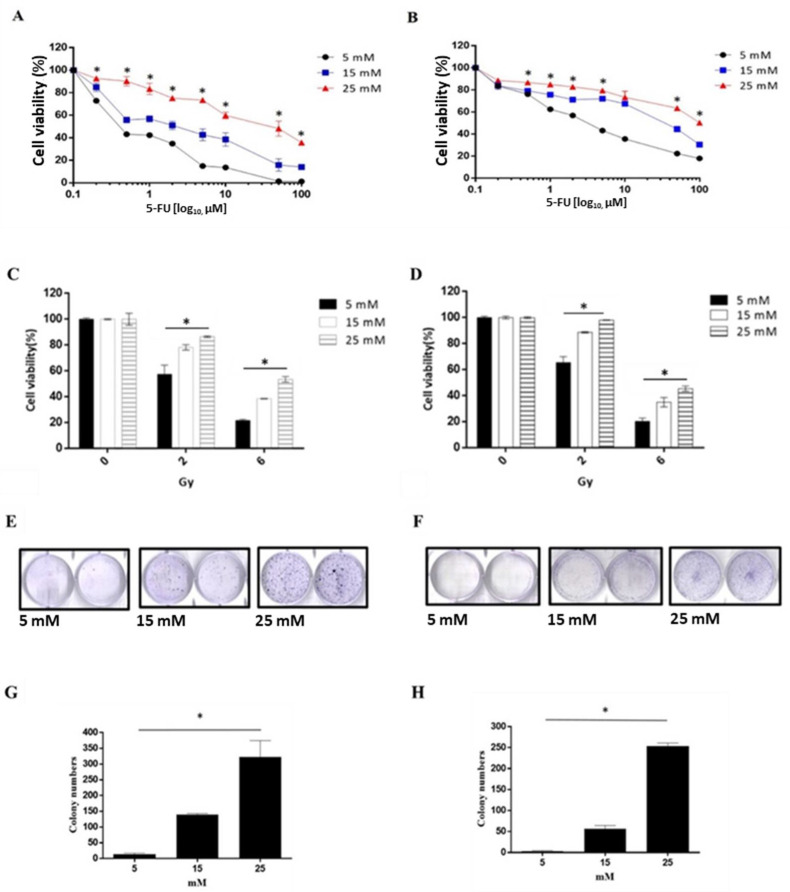
Effects of high glucose on chemoresistance and radioresistance in rectal cancers. HCT116 (**A**) and SW480 (**B**) cells were treated with serial dilution of 5-FU in different glucose concentrations for 48 h and analyzed by measuring relative luminescence units (RLU). HCT116 (**C**) and SW480 (**D**) cells were exposed to 0, 2, or 6 Gy in 5, 15, or 25 mM glucose concentrations for 48 h. The cell viability was calculated by measuring RLU. In the colony formation analysis, HCT116 (**E**) and SW480 (**F**) cells were cultured in the media with 5, 15, or 25 mM glucose concentration exposed to 2 Gy. After 2 weeks, cells were stained and colony formation was calculated. The colony numbers were evaluated (**G**,**H**). Cell viability (%) = (abs sample − abs blank)/(abs control − abs blank) × 100. Data are presented as mean  ±  SD (*n*  =  3; * *p* < 0.05).

**Figure 4 cancers-14-04053-f004:**
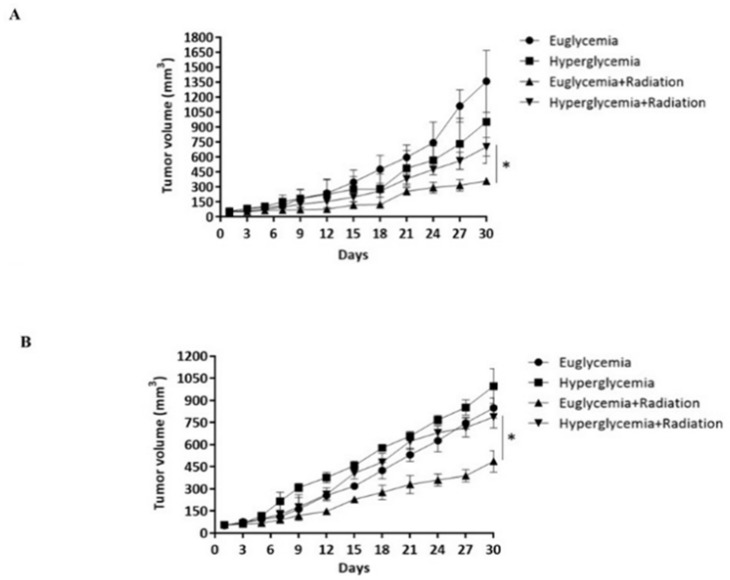
Effects of hyperglycemia promote the radioresistance of rectal cancer through overexpression of HIF-1α. Establishment of HCT116 and SW480 for ectopic rectal cancer with euglycemia or hyperglycemia, respectively. After receiving radiotherapy or not three times per week, the tumor volumes of all groups were measured every two days until day 30. (HCT116, (**A**); SW480, (**B**)) Rectal cancer masses were removed and collected subcutaneously after the measured endpoint (day 30) from the euglycemic or hyperglycemic mice. The HIF-1α level and apoptosis of cancer cells were detected by anti-HIF-1α antibody or Tunel assay in HCT116 (**C**) or SW480 (**D**). HIF-1α protein was stained in the cytoplasm (brown color) and Tunel was stained in the nuclear tumor cells. Tunel staining statistics (**E**,**F**) in HCT116 and SW480, respectively. Data are presented as mean  ±  SD (*n*  =  4; * *p* < 0.05).

**Figure 5 cancers-14-04053-f005:**
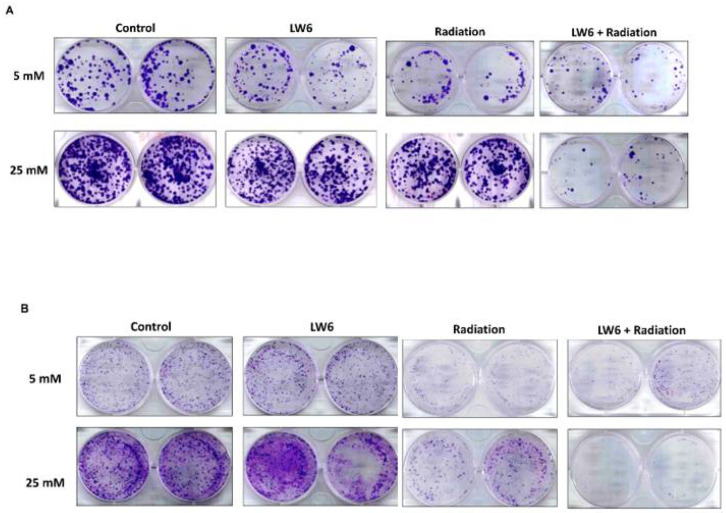
HIF-1α inhibitor increases the cytotoxic effect of radiation on rectal cancer cell lines in a high glucose environment. Rectal cancer cells were cultured in a high glucose concentration medium after being treated with/without radiation or LW6 for three weeks and the colony formation of HCT116 (**A**) or SW480 (**B**) is shown. The statistics of colony formation are shown (**A**,**B** bottom). The cell plates were collected. To detect HIF-1α level and β-actin level by Western blotting (HCT116, (**C**); SW480, (**D**)). The HIF-1α levels/β-actin ratio statistics in HCT116 (**E**) and SW480 (**F**), respectively. Data are presented as mean  ±  SD (*n*  =  3; * *p* < 0.05).

**Figure 6 cancers-14-04053-f006:**
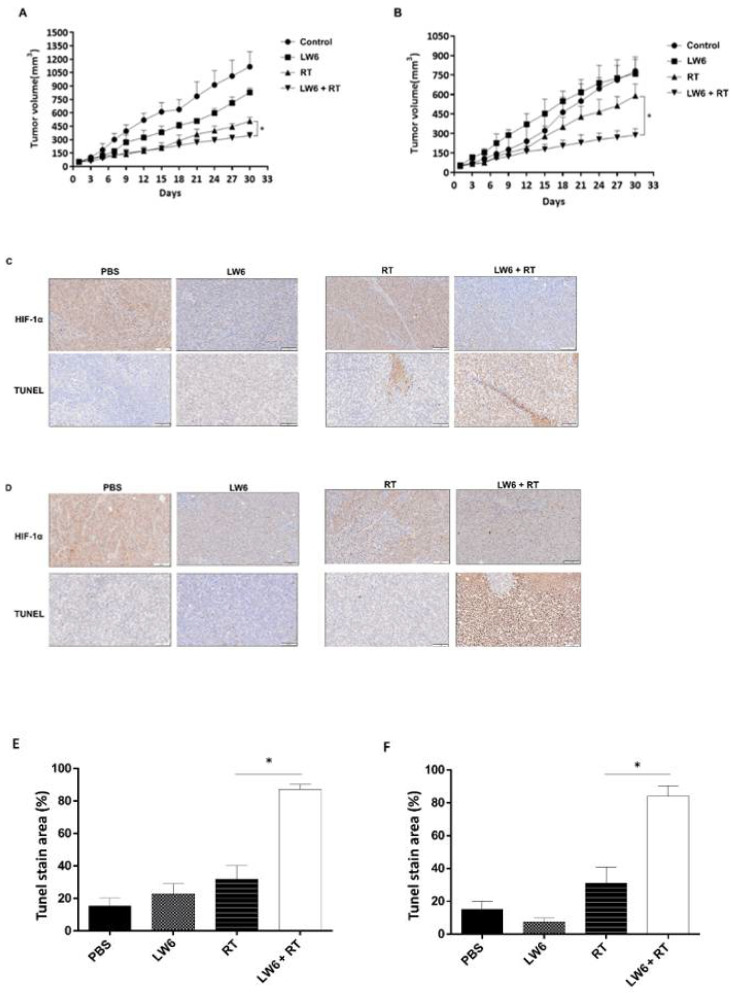
Analysis of the effect of combined LW6 on the radiation of hyperglycemic rectal cancer. Hyperglycemic ectopic rectal cancer was treated with LW6, radiation, or combined LW6 with radiation. The tumor volume was measured every two days until day 30 (**A**,**B**). Rectal cancer masses were removed and collected subcutaneously after the measured endpoint, immunohistochemically analysis of HIF-1α expression, and Tunel assay in HCT116 (**C**) or SW480 (**D**). Tunel staining statistics (**E**,**F**) in HCT116 and SW480, respectively. Data are presented as mean  ±  SD (*n*  =  4; * *p* < 0.05).

**Table 1 cancers-14-04053-t001:** Characteristics of 54 rectal cancer patients receiving preoperative CCRT and association between HbA1c level with the CCRT efficacy. The Chi-square test was used for statistical analysis. The statistical significance level was *, *p* < 0.05.

Characteristics			HbA1C ≤ 6.5%		HbA1C > 6.5%		*p*
Total	N	%	*n* = 41	(%)	*n* = 13	(%)	
**Age (year)**							0.534
<65	37	68.5	29	70.7	8	61.5	
≥65	17	31.5	12	29.3	5	38.5	
**Gender**							0.822
Male	36	66.7	27	65.9	9	69.2	
Female	18	33.3	14	34.1	4	30.8	
**Clinical tumor depth (cT)**							0.653
T3	48	88.9	36	87.8	12	92.3	
T4	6	11.1	5	12.2	1	7.7	
**Clinical lymph node metastasis (cN)**							0.536
N0	12	22.2	10	24.4	2	15.4	
N1	26	48.1	18	43.9	8	61.5	
N2	16	29.6	13	31.7	3	23.1	
**Clinical tumor disease Stage**							0.496
II	12	22.2	10	24.4	2	15.4	
III	42	77.8	31	75.6	11	84.6	
**Pathologic tumor depth (ypT)**							
ypT0	16	29.6	10	24.4	6	46.2	0.338
ypT1	6	11.1	5	12.2	1	7.7	
ypT2	17	31.5	12	29.3	5	38.4	
ypT3	15	27.8	14	34.1	1	7.7	
**Pathologic lymph node metastasis (ypTN)**							0.457
ypN0	44	81.5	32	78	12	92.3	
ypN1	7	13.0	6	14.6	1	7.7	
ypN2	3	5.5	3	7.3	0	0	
**Pathologic tumor disease stage**							0.012 *
pCR	15	27.8	14	34.1	1	7.7	
I	20	37.0	11	26.8	9	69.2	
II	9	16.7	6	14.6	3	23	
III	10	18.5	10	24.4	0	0	
**Lymphovascular invasion**							0.148
Yes	12	22.2	11	26.8	1	7.7	
No	42	77.8	30	73.2	12	92.3	
**Perineural invasion**							0.242
Yes	4	7.4	4	9.8	0	0	
No	50	92.6	37	90.2	13	100	
**Tumor regression score**							0.017 *
0 (CR)	16	29.6	15	36.6	1	7.7	
1	25	46.3	14	34.1	11	84.6	
2	10	18.5	9	22	1	7.7	
3	3	5.6	3	7.3	0	0	
**Pathologic complete response**							0.047 *
Yes	16	29.6	15	36.6	1	7.7	
No	38	70.4	26	63.4	12	92.3	

**Table 2 cancers-14-04053-t002:** Association between pre-CCRT HIF-1α expression with clinicopathological parameters in 54 rectal cancer patients receiving preoperative CCRT. The Chi-square test was used for statistical analysis. The statistical significance level was *, *p* < 0.05.

Characteristics			HIF1-α Expression				*p*
			Low		High		
**Total**	N	%	37	68.5	17	31.5	0.298
**Age (Year)**							
<65	37	68.5	27	73	10	58.8	
≥65	17	31.5	10	27	7	41.2	
**Gender**							0.407
Male	36	66.7	26	70	10	41.2	
Female	18	33.3	11	30	7	58.8	
**Clinical Tumor Depth (cT)**							0.078
T3	48	88.9	31	83.8	17	100	
T4	6	11.1	6	16.2	0	0	
**Clinical Lymph Node Metastasis (cN)**							
N0	12	22.2	10	27	2	11.8	0.081
N1	26	48.1	14	37.8	12	70.6	
N2	16	29.6	13	35.1	3	17.6	
**Clinical Tumor Disease Stage**							0.21
II	12	22.2	10	27	2	11.8	
III	42	77.8	27	73	15	88.2	
**Pathologic Tumor Depth (ypT)**							0.692
ypT0	16	29.6	12	32.4	4	23.5	
ypT1	6	11.1	5	13.5	1	5.9	
ypT2	17	31.5	10	27	7	58.8	
ypT3	15	27.8	10	27	5	29.4	
**Pathologic Lymph Node Metastasis (ypTN)**							0.778
ypN0	44	81.5	31	83.8	13	76.5	
ypN1	7	13.0	4	8.1	3	17.6	
ypN2	3	5.5	2	8.1	1	5.9	
**Pathologic Tumor Disease Stage**							
pCR	15	27.8	14	37.8	1	5.9	0.083
I	20	37.0	13	35.1	7	41.2	
II	9	16.7	5	13.5	4	23.5	
III	10	18.5	5	13.5	5	29.4	
**Lymphovascular Invasion**							0.009 *
Yes	12	22.2	0	0	3	17.6	
No	42	77.8	37	100	14	82.4	
**Perineural Invasion**							0.002 *
Yes	4	7.4	0	0	4	23.5	
No	50	92.6	37	100	13	76.5	
**Tumor Regression Score**							<0.001 *
0 (CR)	16	29.6	15	40.5	1	5.9	
1	25	46.3	19	51.4	6	35.3	
2	10	18.5	3	8.1	7	41.2	
3	3	5.6	0	0	3	17.6	
**Pathologic Complete Response**							0.01 *
Yes	16	29.6	15	40.5	1	5.9	
No	38	70.4	22	59.5	16	94.1	
**HbA1c**							0.046 *
≤6.4	37	68.5	31	83.8	10	58.8	
>6.5	17	31.5	6	16.2	7	41.2	

## Data Availability

The data presented in this study are available on request from the corresponding author. The data are not publicly available due to involvement consent privacy.

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
