# Peer review of "HIF-1α Expression Increases Preoperative Concurrent Chemoradiotherapy Resistance in Hyperglycemic Rectal Cancer"

_cancers, 2022, doi:10.3390/cancers14164053_

Round 1

Reviewer 1 Report

Reviewer’s comment:

The article entitled: “HIF-1α expression increases preoperative concurrent  chemoradiotherapy resistance in hyperglycemic rectal cancer” had demonstrated that hyperglycemia induces the expression of HIF-1α and cause CCRT tolerance in rectal cancer. The authors suggest that combining HIF-1α inhibitors could reverse radioresistance in a high glucose environment. Their method, material result are clear but there are some issues that should be dressed before the article could be considered for acceptance

1)      The author should describe the diabetic history of all their patients, such as how they receive anti-diabetic drugs and how long had they had been treated. This is important to clarify if well-treated diabetic patients are the same outcome compared with non-diabetic patient.

2)      As we known, the treatment modalities and pathomechanism between rectal cancer and colon cancer are very different. Due to all the patients are rectal cancer patient, the author should  describe more in the discussion part about their findings should be used in rectal cancer or all colon-rectal cancers.

3)      The author used mice  as an animal model to confirm that HIF-1 alpha inhibitor LW6 increases the efficacy of radiotherapy for hyperglycemia ectopic colorectal cancer by  inhibiting HIF-1α expression. It would be more convincible if they could put some mice tumor change pictures if could.

Author Response

Thanks to Reviewer 1's suggestion. our reply and manuscript revisions are all in the attachment (because the file size is too large, attach the Google Drive link:
https://drive.google.com/drive/folders/1jfgJxZhASlmRU1H-ha5pzpBjTyqM4eQJ?usp=sharing)

Reviewer 2 Report

This present article by Huang et al discusses how hypoglycemia affects radiochemotherapy in colorectal cancer? Moreover, correlating hypoxia-inducible factor-1 alpha (HIF-1α) in rectal cancer patients with preoperative CCRT. I am not accepting this work for publication in its current form. Here, I included a few suggestions to improve this article  for publication.

Major: 

  1.  Figure 1, quantification is missing, scale bars need to be included. 

  2. Figure 2, quantification was performed from a single experiment, the experiment must be repeated for three times and error bars need to be included. The present quantification mentioned for the blots need to be recalculated. Example, OGT, HIF-1α normalization is not reliable. 

  3. Figure 4, Tunel assay quantification is missing. However, they indicated in the figure legend. 

  4. Figure 5, panel A, B the figure size needs to be resized because it lacks clarity. Panel C & D the experiment must be repeated for three times and error bars need to be included. 

  5. Figure 6, the picture resolution is not reliable as Figure 4. Tunel assay quantification is missing. However, they indicated in the figure legend.

Author Response

Thanks to Reviewer 2's suggestion. our reply and manuscript revisions are all in the attachment (because the file size is too large, attach the Google Drive link:
https://drive.google.com/drive/folders/1jfgJxZhASlmRU1H-ha5pzpBjTyqM4eQJ?usp=sharing)

Reviewer 3 Report

This is interesting paper. The authors confirmed that a high glucose environment could induce the GLUT1-OGT-HIF-1α signaling pathway to promote 5FU-resistance and radioresistance of CRC cells by overexpressing HIF-1α. Moreover, they showed that the HIF-1α inhibitor, LW6, could increase the efficacy of radiotherapy for hyperglycemia ectopic CRC by inhibiting HIF-1α expression. The description and figures demonstrating in vitro experiment and experiment on ectopic CRC are clearly presenrted. 

The clinical part of the article should be more precisely described:

1) The authors stated that clinical stage of rectal cancer was determined by means of computed tomography. Why MRI was not performed as it is nowadays the recommended diagnostic tool for rectal cancer?

2) Was mesorectal fascia involvement in preoperative imaging (MRF+) taken into consideration when decision was made on preoperative treatment?

3) Are patients with rectal cancer qualified to other preoperative treatment than CCRT - i.e. radiotherapy 5x5 Gy or total neoadjuvant treatment. If radiotherapy 5x5 Gy is used what factor decides which type of neoadjuvent treatment the patient receives?

4) What statistical test was used to compare groups of patients in Table 1. The authors mention Student t-test but for this test normal distribution should be confirmed. What test was used to confirm normal distribution of variables? Usually the variables compared in Table 1 have not normal distribution so different statistical tests should be implemented.

5) Table 2 - title of the table includes the word "correlation" but no correlation coefficients were examined. The table compares clinical variables between patients with low and high HIF-1alfa expression. Once again statistical test should be mentioned below the table. Normality of distribution of analysed variables should be comnfirmed. 

6) Time interval between the end or CCRT and surgery should be specified. 

7) Due to the low number of participants (13 with HbA1c> 6,5 and 41 with HbA1c>=6,5) the study is probably underpowered. It should be added as limitation of the study at the end of the discussion. 

8) English language should be corrected - especially in the last paragraph of the results section.

Author Response

Thanks to Reviewer 3's suggestion. our reply and manuscript revisions are all in the attachment (because the file size is too large, attach the Google Drive link:
https://drive.google.com/drive/folders/1jfgJxZhASlmRU1H-ha5pzpBjTyqM4eQJ?usp=sharing)

Round 2

Reviewer 2 Report

This revised version substantially improves the quality of their research work. The Authors tried their best to answer all reviewers comments and provided new data. Therefore, I accept this work for publication.